# Adolescent well-being and learning in times of COVID-19—A multi-country study of basic psychological need satisfaction, learning behavior, and the mediating roles of positive emotion and intrinsic motivation

Julia Holzer[1]*, Selma Korlat[1], Christian Haider[1], Martin Mayerhofer[2], Elisabeth Pelikan[1], Barbara Schober[1], Christiane Spiel[1], Toumazis Toumazi[3], Katariina Salmela-Aro[4], Udo Käser[5], Anja Schultze-Krumbholz[6], Sebastian Wachs[7,8], Mukul Dabas[9], Suman Verma[10], Dean Iliev[11], Daniela Andonovska-Trajkovska[11], Piotr Plichta[12], Jacek Pyżalski[13], Natalia Walter[13], Justyna Michałek-Kwiecień[14], Aleksandra Lewandowska-Walter[14], Michelle F. Wright[15,16], Marko Lüftenegger[1,17]

**1** Department of Developmental and Educational Psychology, Faculty of Psychology, University of Vienna, Vienna, Austria, **2** Department of Mathematics, Faculty of Mathematics, University of Vienna, Vienna, Austria, **3** Cyprus Computer Society, Nicosia, Cyprus, **4** Department of Educational Sciences, University of Helsinki, Helsinki, Finland, **5** Department of Psychology, University of Bonn, Bonn, Germany, **6** Technische Universität Berlin, Chair of Educational Psychology, Berlin, Germany, **7** Department of Educational Studies, University of Potsdam, Potsdam, Germany, **8** National Anti-Bullying Research and Resource Centre, Dublin City University, Dublin, Ireland, **9** Coursera India, Gurgaon, India, **10** Department of Human Development & Family Relations, Government Home Science College, Panjab University, Chandigarh, India, **11** Faculty of Education, St. Kliment Ohridski University, Bitola, North Macedonia, **12** Faculty of Historical and Pedagogical Sciences, University of Wrocław, Wrocław, Poland, **13** Faculty of Educational Studies, Adam Mickiewicz University, Poznań, Poland, **14** Institute of Psychology, Faculty of Social Science, University of Gdańsk, Gdańsk, Poland, **15** Department of Psychology, Penn State University, University Park, Pennsylvania, United States of America, **16** Faculty of Social Studies, Masaryk University, Brno, Czech Republic, **17** Department for Teacher Education, Centre for Teacher Education, University of Vienna, Vienna, Austria

* julia.holzer@univie.ac.at

**Data Availability Statement:** The complete set of items as well as the data related to the present

## Abstract

The sudden switch to distance education to contain the outbreak of the COVID-19 pandemic has fundamentally altered adolescents' lives around the globe. The present research aims to identify psychological characteristics that relate to adolescents' well-being in terms of positive emotion and intrinsic learning motivation, and key characteristics of their learning behavior in a situation of unplanned, involuntary distance education. Following Self-Determination Theory, experienced competence, autonomy, and relatedness were assumed to relate to active learning behavior (i.e., engagement and persistence), and negatively relate to passive learning behavior (i.e., procrastination), mediated via positive emotion and intrinsic learning motivation. Data were collected via online questionnaires in altogether eight countries from Europe, Asia, and North America (N = 25,305) and comparable results across countries were expected. Experienced competence was consistently found to relate to positive emotion and intrinsic learning motivation, and, in turn, active learning behavior in terms of engagement and persistence. The study results further highlight the role of perceived relatedness for positive emotion. The high proportions of explained variance speak

study are available in the Open Science Framework repository (DOI: 10.17605/OSF.IO/X89BZ).

**Funding:** This work was funded by the Vienna Science and Technology Fund (WWTF) [https://www.wwtf.at/] and the MEGA Bildungsstiftung [https://www.megabildung.at/] through project COV20-025, as well as the Academy of Finland [https://www.aka.fi] through project 1308351 and 1336138. BS is the grant recipient of COV20-025. KSA is the grant recipient of 1308351 and 1336138. Open access funding was provided by University of Vienna. The funders had no role in study design, data collection and analysis, decision to publish, or preparation of the manuscript.

**Competing interests:** The authors have declared that no competing interests exist.

in favor of taking these central results into account when designing distance education in times of COVID-19.

## Introduction

The COVID-19 pandemic has drastically changed everyday lives around the globe within a short period of time. To curb the spread of the virus, various containment measures were instituted. These included school closures and a switch to distance education, affecting hundreds of millions of learners [1]. Subsequently, students worldwide have been facing a fundamentally altered situation with respect to schooling and their lives as a whole. Consistent with a large body of evidence on the high secondary costs of school closures [2, 3], studies undertaken in times of COVID-19 similarly point to a great risk for adolescents' well-being and positive development. These include a greater risk of anxiety and depression [4, 5] and a lack of resources and services usually provided by schools, such as counseling, psychological services, or special education [6]. Warnings of long-term consequences for the affected cohorts were equally issued by scientists from economic, educational, and social sciences [7–9]. Given these risks and challenges, the present research aims to identify characteristics that relate to adolescents' psychological well-being in terms of positive emotion and intrinsic learning motivation, as well as qualities of their learning process during unplanned and involuntary distance education. Following Self-Determination Theory (SDT; [10, 11]), we focus on the role of basic psychological need satisfaction for the aforementioned emotional, motivational, and behavioral outcomes and investigate whether core postulates of SDT remain valid during the exceptional situation of COVID-19. To take a broad perspective and to derive common conclusions applicable across different cultural contexts, data were collected in altogether eight countries from Europe, Asia and North America.

### Adolescent well-being and basic psychological need satisfaction

Adolescent well-being represents a vital resource for positive development and a prerequisite for successful learning. Converging evidence points to its relevance for general health and adaptive cognitive and behavioral outcomes [12, 13]. According to SDT [10, 11], well-being is generally understood as consisting of both hedonic and eudaimonic well-being. While hedonic well-being refers to well-being as an outcome in terms of positive emotions, eudaimonic well-being is characterized as a process of pursuing intrinsic goals and values for their own sake (i.e., intrinsic motivation [14, 15]). SDT states that satisfaction of the basic psychological needs for competence, autonomy, and relatedness represents a universal requirement for the experience of well-being in terms of positive emotion and intrinsic motivation, independent of cultural context, social class, or gender [11, 16]. The need for competence refers to experiencing one's behavior as effective. For example, students experience competence when they are able to master their school-related tasks. Autonomy refers to experiencing one's behavior as volitional and self-endorsed. For instance, students experience autonomy when they deliberately choose to devote time and energy to learning. Finally, relatedness refers to feeling connected and experience mutual support from significant others [10, 17, 18]. A large body of empirical research has accounted for relations between basic need satisfaction and intrinsic motivation, coping, and positive emotions in adults as well as adolescents across different life domains [19–21]. In accordance with SDT´s claim for universality, a growing amount of studies has found evidence for its cross-cultural applicability in different cultures on every continent

[22–26]. Moreover, SDT has repeatedly been applied in the educational context, and beneficial outcomes among students have effectively been targeted [27–29].

## Distance education and basic need satisfaction in times of COVID-19

In order to continue education in the face of COVID-19, schools have provided various remote teaching solutions [30], in this study referred to as distance education. The unplanned and involuntary institution of distance education has posed challenges, but also opportunities for adolescents' learning and basic need satisfaction during the pandemic. One central issue is the physical separation between educators and learners, as well as among peers, leading to limited spontaneous interaction and less informal, personal exchange. As feeling connected to others has consistently been shown to relate to academic success and well-being in traditional as well as distance education settings [31–33], it has been argued that students' personal needs, feelings, and difficulties should be explicitly addressed in distance education settings [34]. Moreover, maintaining social contacts is considered to be even more important for the well-being of adolescents than for adults, as young people predominantly gain meaning from their social relations [35, 36]. This indicates a high relevance of experienced relatedness for adolescents' well-being during the COVID-19 pandemic. However, the fact that students in a distance education setting can approach learning in a highly individualized manner also harbors great opportunities. Distance education may provide learners with manifold opportunities to practice, test, and expand their knowledge at their own pace [37]. This allows to optimally challenge students and to achieve a high degree of individualization of their learning processes [38, 39]. However, individualized learning also requires the learner to be able to handle flexibility and not all learners are able to cope with distance learning equally well [40]. Students in distance learning environments should therefore be encouraged to employ adequate learning strategies with respect to self-regulation [41]. When implemented successfully, individualized, autonomy-supportive learning environments have consistently been shown to create optimal conditions for learners to experience competence [17]. As stated by SDT, both experienced competence and autonomy are necessary conditions to develop and maintain intrinsic motivation and, along with perceived relatedness, contribute to positive affective states. Moreover, SDT presumes that positive emotions and intrinsic motivation lead individuals' behavior towards active engagement and persistence when facing tasks. Inversely, if the basic needs are unsatisfied, intrinsic motivation and positive affect are deprived, which in turn leads to passive behavior [11]. Applied to the educational context, there is empirical evidence in favor of several of these assumptions. Accordingly, studies report positive relations between intrinsic motivation and engagement [42], as well as persistence [43]. Further findings account for associations of positive emotions with learning engagement [44], and persistence [45]. Conversely, procrastination was found to be negatively related to positive affect [46], and intrinsic motivation [47]. In addition, there are studies pointing to direct effects of basic psychological need satisfaction on active learning behavior, respectively negative relations of need satisfaction with passive learning behavior. Accordingly, learning engagement was found to be predicted by experienced competence and relatedness [48] and autonomy-supportive learning environments [49]. Persistence was found to be associated with experienced competence and autonomy [50, 51] and procrastination was found to be negatively associated to autonomy-supportive teaching [52], and experienced autonomy [53].

The COVID-19 pandemic as well as the undertaken containment measures represent a potentially need thwarting context, posing a risk to basic need satisfaction [54, 55]. Need thwarting contexts are well researched on different individual levels like parenting styles [56] or meso levels like school contexts [57], widely supporting SDT´s claims. To the best of our

knowledge, no prior research has been conducted to explore SDT in the context of past epidemics (e.g., SARS). However, there is now a growing body of studies conducted during the COVID-19 pandemic across different cultures [54, 58, 59], pointing to the universal applicability of SDT even in exceptional situations like a pandemic. In accordance with these results and SDT´s claim for universality [11, 16], it seems reasonable to assume basic principles of STD to be valid in a situation of unplanned distance education during the COVID-19 pandemic.

## The present research

In light of the major challenges of COVID-19 for adolescents' positive development, the present research aims to identify psychological characteristics that relate to adolescents' well-being in terms of positive emotion and intrinsic learning motivation [14, 15], and key characteristics of their learning behavior. Following SDT, we assume that basic need satisfaction relates to positive emotional and motivational outcomes, which, in turn, positively relate to active learning behavior (i.e., engagement and persistence), and negatively relate to passive learning behavior (i.e., procrastination).

**Hypotheses.** As a general model we assume that basic psychological need satisfaction has direct effects on learning behavior as well as indirect effects, mediated via positive emotion and intrinsic learning motivation. Concretely we formulated the following hypotheses:

We assume that all three basic needs (perceived competence, autonomy, and relatedness) will positively predict positive emotion (Hypotheses 1a-c) and intrinsic learning motivation (H 1d-f), and that positive emotion and intrinsic learning motivation will positively predict engagement (H 2a, H 3a) and persistence (H 2b, H 3b), and negatively predict procrastination (H 2c, H 3c). Furthermore, we assume that experienced competence, autonomy, and relatedness will have a positive indirect effect on engagement (H 4a-c) and persistence (H 4d-f), and a negative indirect effect on procrastination (H 4g-i), mediated via positive emotion. Similarly, we assume that experienced competence, autonomy, and relatedness will have a positive indirect effect on engagement (H 5a-c) and persistence (H 5d-f), and a negative indirect effect on procrastination (H 5g-i), mediated via intrinsic learning motivation. Moreover, we assume positive direct effects of perceived competence, autonomy, and relatedness on engagement (H 6a-c), persistence (H 6d-f), and negative direct effects on procrastination (H 6g-i). A conceptual diagram of the assumed relations is depicted in Fig 1.

In view of the existing body of literature and SDT assumptions, the desideratum of this research is to investigate whether previous findings and core postulates of SDT remain valid during the exceptional situation of COVID-19. To further account for a possible universality of the findings, data were collected in altogether eight countries from Europe, Asia and North America, namely Austria, Cyprus, Finland, Germany, India, North Macedonia, Poland and the USA. The presumed relations are examined separately for each of the countries and then summarized with respect to common findings and discrepancies. In accordance with SDT´s claim for universality [11, 16] and a large body of studies accounting for its cross-cultural applicability [22–26], we expect to obtain comparable results across the countries of data collection with respect to the directions and effect sizes (H 7).

In line with previous studies that indicate associations of age with intrinsic learning motivation [60] and positive emotion [61], we treated participants' age as control variables.

## Method

### Sample and procedure

The overall sample comprised 25,305 students (37.9% males, 61.5% females, 0.6% diverse). Their mean age was 14.86 years ($SD$ = 2.43, $Mdn$ = 15.00, $Range$ = 10–21). All students

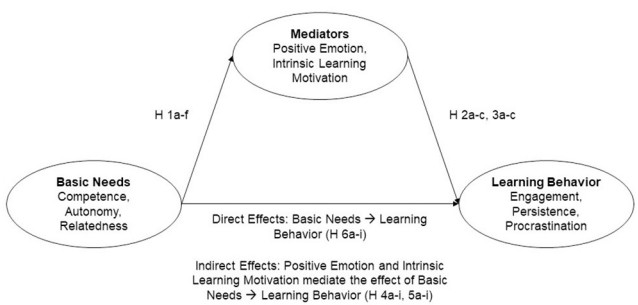

**Fig 1. Overview of the assumed relations (direct and indirect effects) in this study.**

reported attending secondary schools. Their age range reflects the fact that in many countries of data collection, students older than 18 may still attend secondary education e.g., if the school form integrates vocational training [62] or due to grade repetition. Descriptive statistics on the samples for each country are given in Table 1.

The cross-country study was conceived by the research group in Austria. It was carried out as part of a larger research project focused on "Learning under COVID-19 circumstances" [63–66], approved and supported by the Federal Ministry of Education, Science and Research, and was carried out starting with April 2020. To expand the one-country scope, the Austrian group invited cooperation partners from their research networks to participate in a joint study. These networks were the European Family Support Network (COST Action 18123), the Transnational Collaboration on Bullying, Migration and Integration at School Level (COST Action 18115), and the International Panel on Social Progress. Subsequently, researchers in seven further countries joined the project. In a next step, the original German questionnaire was translated into the respective languages using translation-back-translation method [67]. Cooperation partners each ensured that all ethical and legal requirements related to data collection were met in their contexts of data collection, as it was ensured in Austria. All procedures performed in this study were in accordance with 1964 Helsinki declaration and its later amendments or comparable ethical standards.

Data were collected via online questionnaires between April and June 2020. To recruit the sample, in each country the link to the respective online questionnaire was distributed by contacting diverse stakeholders including school boards, educational networks, and school principals. Before answering the items, participants were informed about the study's goals, expected duration of filling in the questionnaire, inclusion criteria for participation (i.e., attending secondary school in the respective country), and the complete anonymity of their data. All

**Table 1. Sociodemographic characteristics of participants per country.**

| Country of data collection | | Gender | | | Age (years) | | | |
|---|---|---|---|---|---|---|---|---|
| | *N* | male | female | diverse | *M* | *SD* | *Mdn* | *Range* |
| Austria | 19,337 | 37.9% | 61.6% | 0.5% | 14.56 | 2.49 | 14.00 | 10–21 |
| Cyprus | 141 | 32.6% | 65.2% | 2.2% | 13.83 | 1.16 | 14.00 | 12–17 |
| Finland | 614 | 23.3% | 74.9% | 1.8% | 17.01 | 0.83 | 17.00 | 14–20 |
| Germany | 629 | 30.4% | 69.1% | 0.5% | 15.73 | 2.76 | 16.00 | 10–21 |
| India | 2,618 | 45.2% | 54.2% | 0.6% | 15.74 | 1.39 | 16.00 | 12–20 |
| North Macedonia | 1,084 | 30.7% | 69.3% | 0.0% | 16.98 | 1.11 | 17.00 | 14–20 |
| Poland | 379 | 36.4% | 62.5% | 1.1% | 14.26 | 2.46 | 14.00 | 10–19 |
| USA | 503 | 45.0% | 53.3% | 1.7% | 14.63 | 2.16 | 15.00 | 10–19 |

students participated voluntarily and only those who gave active consent were included in the dataset. Due to the special circumstances of data collection (i.e., nationwide lockdowns) and the fact that the questionnaire did not deal with any sensitive issues, the Ministry of Education carefully considered the content of the study, its possible benefits, and authorized the study to be conducted without the explicit consent of participants' legal guardians.

During the whole period of data collection in the respective countries, school attendance was either fully suspended or drastically limited (e.g., final year students attended school for a few hours a week), and schools provided distance education via e-learning and distribution of teaching material.

## Measures

In order to target the new situation, existing scales from established measures were slightly adapted and complemented by a small number of newly developed items. The measure was then revised based on expert judgements and piloted with cognitive interview testing among a group of adolescents. To further ensure construct validity of the implemented measures in our study, we conducted confirmatory factor analyses (CFA) of the measurement models and analyzed composite reliabilities of the scales (CR; [68]) separately for each country of data collection.

All items were rated on a 5-point scale ranging from 1 (strongly agree) to 5 (strongly disagree). Participants were instructed to answer the items with respect to the current situation (learning from home due to the coronavirus pandemic). We conducted the analyses with recoded items so that higher values reflected higher agreement with the statements.

*Competence* satisfaction was measured with three items adapted from the Work-related Basic Need Satisfaction Scale (W-BNS; [69]). The work-related items were adapted to the school context (sample item: "These days I am able to successfully complete most of my schoolwork").

*Autonomy* satisfaction was assessed with three newly developed items addressing the extent to which students could approach studying how it suited them best (sample item: "Currently, I can decide on my own how I want to approach studying").

*Relatedness* satisfaction was measured with three items adapted from the connectedness subscale of the EPOCH Measure [70]. Contrary to perceived competence and autonomy, items targeting perceived relatedness did not specifically refer to the school context, but to significant others in general (sample item: "When something good happens to me, I have people who I tell about it").

*Positive emotion* was measured with two items from the Scale of Positive And Negative Experience (SPANE; [71]; "I feel good", "I feel content"), and one item adapted from the EPOCH Measure ([70]; "Even though things are tough right now, I think everything will be okay").

*Intrinsic learning motivation* was assessed with three items adapted from the Scales for the Measurement of Motivational Regulation for Learning in University Students (SMR-LS; [72]; sample item: "Currently, I really enjoy studying and working for school").

*Engagement* was measured with three adapted items of the subscale engagement of the EPOCH Measure ([70]; sample item: "When I am working on my schoolwork, I get completely absorbed in what I am doing").

*Persistence* was measured with three adapted items from the perseverance subscale of the EPOCH Measure ([70]; sample item: "If I start a task for school, I finish it.").

*Procrastination* was measured with three items, adapted from the Procrastination Questionnaire for Students (Prokrastinationsfragebogen für Studierende, PFS; [73]; sample item: "When I am currently studying, I put off tasks until the last minute").

Table 2 accounts for the CR indices per country.

The complete set of items as well as the data related to the present study are publicly available online under doi.org/10.17605/OSF.IO/X89BZ.

## Data analysis

Data were analyzed using SPSS version 25.0 [74] and Mplus version 8.4 [75]. Preliminary analyses to ensure internal consistency (CR; [68]) and construct validity (CFAs) were conducted applying robust maximum likelihood estimation (MLR). Goodness-of-fit was evaluated using $\chi^2$ test of model fit, the comparative fit index (CFI), and the root mean square error of approximation (RMSEA). Following Hu & Bentler [76], CFI > 0.95 and 0.90, and RMSEA < .06 and .08 were considered as cutoff scores accounting for excellent, respectively adequate model fit.

To test for measurement invariance between the countries of data collection, we set up multi-group CFAs, one each for the predictors, the mediators, and the outcomes [77]. As usual for studies with multiple groups [78], we tested for configural, metric, and scalar invariance. If configural invariance is established, the same factor structure is valid for each group. Metric invariance implies that students in each country attribute the same meaning to the latent constructs. If scalar invariance is established, the meaning of the levels of the underlying items is equal in the groups [79]. For group comparisons of structural relationships in structural equation modeling, metric invariance is required [77]. For evaluating the measurement invariance assumptions, we followed Chen [80]. Accordingly, declines in CFI ≥.01 and increases in RMSEA ≥.015 indicate meaningful changes in model fit, making assumptions of measurement invariance not tenable.

For the main analyses, statistical mediation analyses were conducted to investigate effects of basic psychological need satisfaction on positive emotion and intrinsic learning motivation as well as direct and indirect effects of basic need satisfaction on learning characteristics via positive emotion and intrinsic learning motivation. The conceptual model of the assumed relations is depicted in Fig 1. To conduct the mediation analysis, we set up a multi-group model with the country of data collection as the grouping variable. Statistical significance testing was conducted at the .05 level. For testing significance of direct and indirect effects within the mediation models, we used bias-corrected bootstrapping confidence intervals based on 10,000 bootstrap draws and applied maximum likelihood estimation (ML). For the interpretation of results, apart from relying on statistical significance, we focused on the effect sizes of the regression parameters. In doing so, we followed Cohen's recommendations [81], according to which standardized values varying around .10, .30, and greater than .50 represent small, medium, and large effects.

In a final step, we explored common findings and discrepancies across countries and examined whether countries differed from one another in terms of the identified effect sizes. Due to the large number of countries, we aimed for an approach that allows to gain a clear overview of which effects differ from one another, taking into account the different statistical power of the respective samples. We therefore used the bootstrapping confidence intervals of the direct and indirect effects to examine whether countries differed from one another in terms of the identified effect sizes. Accordingly, if intervals overlap, i.e., enclose a common range of values, they indicate that the difference between countries is not statistically significant. If there is no overlap, the difference is statistically significant.

**Table 2. Composite reliabilities of the scales per country.**

| Country of data collection | 1 | 2 | 3 | 4 | 5 | 6 | 7 | 8 |
|---|---|---|---|---|---|---|---|---|
| Austria | .85 | .76 | .83 | .86 | .92 | .74 | .74 | .84 |
| Cyprus | .85 | .78 | .81 | .83 | .90 | .82 | .77 | .75 |
| Finland | .85 | .83 | .83 | .86 | .86 | .75 | .76 | .89 |
| Germany | .85 | .75 | .83 | .85 | .89 | .78 | .75 | .84 |
| India | .84 | .83 | .79 | .69 | .87 | .66 | .81 | .61 |
| North Macedonia | .82 | .84 | .87 | .81 | .91 | .73 | .79 | .79 |
| Poland | - * | .84 | .86 | .81 | .89 | .76 | .71 | .80 |
| USA | .82 | .81 | .75 | .82 | .81 | .88 | .82 | .80 |

*Note.* Composite reliabilities accounting for internal consistencies of the scales: 1. Competence, 2. Autonomy, 3. Relatedness, 4. Positive emotion, 5. Intrinsic learning motivation, 6. Engagement, 7. Persistence, 8. Procrastination.

* Due to technical issues, participants in Poland were only presented two items of the competence scale. It was therefore not possible to analyze composite reliability for this scale in this country.

# Results

## Preliminary analyses

Bivariate latent correlations among all variables for each country of data collection are presented in (S1 Appendix). (S2 Appendix) provides descriptive statistics of the scales for each country. The results of the CFAs revealed adequate to excellent fit indices of the measurement model in all countries (see Table 3).

Confirmatory factor analysis for measurement invariance showed excellent model fit for all configural and metric invariance models, and adequate fit for the scalar invariance models. In line with Chen's recommendations [80], there was no substantial difference in model fit regarding CFI and RMSEA between the configural und metric invariance models. Thus, configural and metric invariance can be assumed for all scales. As the scalar invariance assumptions did not hold, indicating that the meanings of the levels of the items were not equal between groups, factor means should not be compared across the countries of data collection. However, as metric invariance was established, group comparisons of the structural relationships of the constructs are legitimate [77]. The results of the measurement invariance testing are reported in Table 4.

**Table 3. Model fit indices of the measurement models for each country of data collection.**

| Country of data collection | $\chi^2$ (df) | $p$ | RMSEA | CFI |
|---|---|---|---|---|
| Austria | 7495.54 (224) | < .001 | .04 | .96 |
| Cyprus | 356.34 (224) | < .001 | .07 | .91 |
| Finland | 605.63 (224) | < .001 | .05 | .95 |
| Germany | 545.78 (224) | < .001 | .05 | .95 |
| India | 1500.48 (224) | < .001 | .05 | .93 |
| North Macedonia | 878.16 (224) | < .001 | .05 | .93 |
| Poland | 467.21 (202) | < .001 | .07 | .92 |
| USA | 653.63 (224) | < .001 | .06 | .91 |

*Note.* CFAs applying robust maximum likelihood estimation (MLR) were conducted to assess construct validity of the measurement model for each country. Goodness-of-fit was evaluated using $\chi^2$ test of model fit, the comparative fit index (CFI), and the root mean square error of approximation (RMSEA). CFI > 0.95 and 0.90, and RMSEA < .06 and.08 were considered as cutoff scores accounting for excellent, respectively adequate model fit.

**Table 4. Measurement invariance testing across countries for the confirmatory factor analytic measurement models for the predictors, mediators, and the outcomes.**

| Models | $\chi^2$ (df) | $p$ | RMSEA | Δ RMSEA | CFI | Δ CFI |
|---|---|---|---|---|---|---|
| **Predictors* (competence, autonomy, relatedness)** | | | | | | |
| Configural | 1532.33 (168) | < .001 | .048 | | .980 | |
| Metric | 1901.05 (204) | < .001 | .048 | 0.000 | .975 | -0.005 |
| Scalar | 4244.60 (240) | < .001 | .069 | 0.021 | .941 | -0.034 |
| **Mediators (positive emotion, intrinsic learning motivation)** | | | | | | |
| Configural | 739.87 (64) | < .001 | .058 | | .990 | |
| Metric | 1066.83 (92) | < .001 | .058 | 0.000 | .985 | -0.005 |
| Scalar | 2110.46 (120) | < .001 | .072 | 0.014 | .970 | -0.005 |
| **Outcomes (engagement, persistence, procrastination)** | | | | | | |
| Configural | 2495.57 (192) | < .001 | .062 | | .963 | |
| Metric | 3203.03 (234) | < .001 | .063 | 0.001 | .953 | -0.010 |
| Scalar | 5203.24 (276) | < .001 | .075 | 0.012 | .922 | -0.031 |

*Note*. $\chi^2$ = chi square test of model fit; CFI = comparative fit index; ΔCFI = change in CFI compared to the weaker measurement invariance model above; RMSEA = root mean square error of approximation; ΔRMSEA = change in RMSEA compared to the weaker measurement invariance model above.

*As participants in Poland, due to technical issues, were only presented two items of the competence scale, Polish data were not included in the measurement invariance analysis for the predictors.

## Main analyses

The results of the multi-group mediation analyses (i.e., direct and indirect effects, and explained variance of dependent variables) for each country of data collection are depicted in Table 5. The bias-corrected bootstrapping confidence intervals for each of the effects are presented in (S3 Appendix).

**Direct effects of basic needs on the mediators (H 1a-f).** In line with Hypothesis 1a, the effect of experienced competence on positive emotion was statistically significant in all countries of data collection except for Cyprus with medium to large effect sizes ranging from $\beta$ = .20 to.61. Positive effects of experienced autonomy on positive emotion (H 1b) were detected in Austria, Finland, India, and North Macedonia with small to medium effect sizes ($\beta$ = .10 to.23). The positive effect of experienced relatedness on positive emotion (H 1c) was statistically significant in all countries except for Cyprus with small to medium effect sizes ($\beta$ = .16 to.43).

Consistent with Hypothesis 1d, the effect of perceived competence on intrinsic learning motivation was statistically significant in all countries with medium to large effect sizes ($\beta$ = .31 to.72). Perceived autonomy statistically significantly predicted intrinsic learning motivation (H 1e) in all countries except Cyprus and the USA with small to medium effects ($\beta$ = .11 to.31). Contrary to Hypothesis 1f, experienced relatedness did not positively predict intrinsic learning motivation, but negatively predicted intrinsic learning motivation in Austria, Germany, and North Macedonia with small effect sizes respectively ($\beta$ = -.05 to -.15). In line with the hypothesis, perceived relatedness positively predicted intrinsic learning motivation in Poland and the USA, with small effects respectively ($\beta$ = .14).

**Direct effects of the mediators on learning behavior (H 2a-c, 3a-c).** Contrary to Hypothesis 2a, engagement was negatively predicted by positive emotion in Austria ($\beta$ = -.13), and Germany ($\beta$ = -.22) with a small and a medium effect. There were no statistically significant associations of engagement and positive emotion in other countries of data collection. Also, in contrast to Hypothesis 2b, positive emotion was a significant negative predictor of

**Table 5. Results of mediation analyses, i.e., direct and indirect effects, and explained variance by the models for each country of data collection.**

| | Austria | | Cyprus | | Finland | | Germany | | India | | North Macedonia | | Poland* | | USA | |
|---|---|---|---|---|---|---|---|---|---|---|---|---|---|---|---|---|
| | Est. (SE) | Std. Est. | Est. (SE) | Std. Est. | Est. (SE) | Std. Est. | Est. (SE) | Std. Est. | Est. (SE) | Std. Est. | Est. (SE) | Std. Est. | Est. (SE) | Std. Est. | Est. (SE) | Std. Est. |
| **Direct Effects** | | | | | | | | | | | | | | | | |
| Positive Emotion → Engagement | **-0.08 (0.01)** | **-0.13** | -0.03 (0.08) | -0.05 | -0.09 (0.07) | -0.13 | **-0.14 (0.04)** | **-0.22** | -0.04 (0.03) | -0.05 | -0.03 (0.04) | -0.05 | -0.05 (0.05) | -0.09 | 0.02 (0.04) | 0.04 |
| Positive Emotion → Persistence | **-0.11 (0.01)** | **-0.16** | -0.12 (0.07) | -0.17 | -0.12 (0.08) | -0.14 | **-0.15 (0.05)** | **-0.21** | -0.05 (0.04) | -0.05 | **-0.08 (0.04)** | **-0.11** | **-0.12 (0.04)** | **-0.22** | 0.09 (0.05) | 0.14 |
| Positive Emotion → Procrastination | **0.12 (0.02)** | **0.11** | 0.07 (0.10) | 0.09 | 0.15 (0.11) | 0.11 | 0.08 (0.08) | 0.07 | 0.05 (0.05) | 0.05 | 0.07 (0.06) | 0.06 | **0.15 (0.07)** | **0.15** | -0.04 (0.06) | -0.05 |
| Learning Motivation → Engagement | **0.25 (0.01)** | **0.45** | 0.10 (0.10) | 0.15 | **0.28 (0.04)** | **0.46** | **0.29 (0.04)** | **0.47** | **0.13 (0.03)** | **0.25** | **0.14 (0.04)** | **0.26** | **0.36 (0.07)** | **0.60** | -0.08 (0.04) | -0.11 |
| Learning Motivation → Persistence | **0.17 (0.01)** | **0.30** | **0.25 (0.09)** | **0.35** | **0.18 (0.06)** | **0.24** | **0.15 (0.05)** | **0.22** | **0.12 (0.03)** | **0.17** | **0.20 (0.04)** | **0.32** | **0.27 (0.05)** | **0.41** | -0.01 (0.05) | -0.01 |
| Learning Motivation → Procrastination | **-0.16 (0.01)** | **-0.16** | 0.02 (0.11) | 0.02 | -0.16 (0.08) | -0.13 | **-0.20 (0.08)** | **-0.18** | **0.09 (0.04)** | **0.13** | -0.06 (0.06) | -0.06 | **-0.36 (0.10)** | **-0.31** | 0.02 (0.06) | 0.02 |
| Competence → Engagement | **0.18 (0.01)** | **0.24** | 0.04 (0.14) | 0.06 | 0.12 (0.07) | 0.21 | **0.28 (0.06)** | **0.38** | **0.19 (0.04)** | **0.31** | 0.05 (0.06) | 0.07 | -0.04 (0.07) | -0.06 | 0.01 (0.05) | 0.01 |
| Competence → Persistence | **0.39 (0.01)** | **0.49** | **0.45 (0.14)** | **0.56** | **0.58 (0.08)** | **0.79** | **0.49 (0.07)** | **0.61** | **0.41 (0.05)** | **0.51** | **0.35 (0.07)** | **0.42** | **0.23 (0.08)** | **0.34** | **-0.13 (0.06)** | **-0.18** |
| Competence → Procrastination | **-0.68 (0.02)** | **-0.50** | **-0.39 (0.15)** | **-0.46** | **-0.94 (0.11)** | **-0.78** | **-0.59 (0.11)** | **-0.44** | -0.09 (0.06) | -0.11 | **-0.52 (0.10)** | **-0.42** | **-0.43 (0.10)** | **-0.36** | -0.05 (0.07) | -0.06 |
| Competence → Positive Emotion | **0.62 (0.01)** | **0.53** | 0.24 (0.18) | 0.21 | **0.54 (0.04)** | **0.61** | **0.59 (0.07)** | **0.51** | **0.30 (0.04)** | **0.37** | **0.33 (0.07)** | **0.29** | **0.55 (0.09)** | **0.45** | **0.21 (0.07)** | **0.20** |
| Competence → Learning Motivation | **0.81 (0.01)** | **0.60** | **0.47 (0.15)** | **0.42** | **0.65 (0.05)** | **0.67** | **0.79 (0.06)** | **0.67** | **0.84 (0.05)** | **0.72** | **0.89 (0.08)** | **0.66** | **0.32 (0.07)** | **0.31** | **0.34 (0.06)** | **0.37** |
| Autonomy → Engagement | 0.02 (0.01) | 0.02 | 0.23 (0.14) | 0.26 | 0.04 (0.04) | 0.06 | -0.04 (0.04) | -0.06 | **0.08 (0.04)** | **0.12** | 0.02 (0.04) | 0.03 | -0.09 (0.05) | -0.13 | 0.04 (0.05) | 0.04 |
| Autonomy → Persistence | -0.02 (0.01) | -0.02 | 0.02 (0.13) | 0.03 | -0.13 (0.04) | -0.16 | 0.03 (0.04) | 0.04 | **0.13 (0.04)** | **0.14** | -0.01 (0.05) | -0.01 | -0.06 (0.06) | -0.09 | 0.10 (0.06) | 0.09 |
| Autonomy → Procrastination | **0.06 (0.01)** | **0.04** | 0.14 (0.17) | 0.13 | 0.13 (0.07) | 0.10 | -0.09 (0.07) | -0.07 | **0.13 (0.05)** | **0.14** | **0.16 (0.07)** | **0.13** | 0.02 (0.09) | 0.02 | 0.09 (0.07) | 0.08 |
| Autonomy → Positive Emotion | **0.12 (0.01)** | **0.10** | 0.28 (0.20) | 0.20 | **0.12 (0.04)** | **0.12** | 0.11 (0.06) | 0.09 | **0.11 (0.05)** | **0.11** | **0.26 (0.06)** | **0.23** | -0.06 (0.08) | -0.05 | 0.11 (0.08) | 0.07 |
| Autonomy → Learning Motivation | **0.19 (0.01)** | **0.14** | 0.25 (0.18) | 0.19 | **0.19 (0.05)** | **0.18** | **0.13 (0.06)** | **0.11** | **0.15 (0.06)** | **0.11** | **0.15 (0.07)** | **0.11** | **0.33 (0.08)** | **0.31** | -0.01 (0.07) | -0.01 |
| Relatedness → Engagement | **0.05 (0.01)** | **0.05** | 0.00 (0.13) | 0.00 | 0.05 (0.05) | 0.07 | 0.06 (0.04) | 0.08 | 0.01 (0.02) | 0.02 | 0.00 (0.03) | 0.00 | -0.01 (0.05) | -0.02 | 0.06 (0.05) | 0.10 |
| Relatedness → Persistence | **0.11 (0.01)** | **0.12** | 0.08 (0.12) | 0.08 | 0.04 (0.05) | 0.04 | 0.05 (0.05) | 0.06 | 0.05 (0.03) | 0.06 | **0.14 (0.04)** | **0.17** | 0.10 (0.06) | 0.16 | **0.06 (0.06)** | **0.07** |
| Relatedness → Procrastination | **-0.07 (0.02)** | **-0.04** | -0.23 (0.16) | -0.21 | 0.13 (0.07) | 0.09 | 0.02 (0.08) | 0.01 | -0.06 (0.04) | -0.08 | **-0.12 (0.06)** | **-0.09** | -0.10 (0.08) | -0.09 | -0.02 (0.08) | -0.03 |
| Relatedness → Positive Emotion | **0.29 (0.01)** | **0.22** | 0.24 (0.19) | 0.16 | **0.26 (0.05)** | **0.24** | **0.21 (0.05)** | **0.16** | **0.24 (0.03)** | **0.26** | **0.24 (0.05)** | **0.21** | **0.30 (0.08)** | **0.27** | **0.47 (0.08)** | **0.43** |
| Relatedness → Learning Motivation | **-0.07 (0.01)** | **-0.05** | 0.09 (0.14) | 0.06 | -0.09 (0.06) | -0.08 | **-0.19 (0.06)** | **-0.15** | -0.01 (0.03) | 0.00 | **-0.09 (0.04)** | **-0.07** | 0.14 (0.06) | 0.14 | **0.13 (0.06)** | **0.14** |
| **Indirect Effects** | | | | | | | | | | | | | | | | |
| Competence → Positive Emotion | | | | | | | | | | | | | | | | |
| Engagement | **-0.05 (0.01)** | **-0.07** | -0.01 (0.02) | -0.01 | -0.05 (0.04) | -0.08 | **-0.08 (0.03)** | **-0.11** | -0.01 (0.01) | -0.02 | -0.01 (0.01) | -0.01 | -0.03 (0.03) | -0.04 | 0.01 (0.01) | 0.01 |
| Persistence | **-0.07 (0.01)** | **-0.08** | -0.03 (0.03) | -0.04 | -0.06 (0.05) | -0.08 | **-0.09 (0.03)** | **-0.11** | -0.02 (0.01) | -0.02 | -0.03 (0.01) | -0.03 | **-0.07 (0.03)** | **-0.10** | 0.02 (0.01) | 0.03 |

*(Continued)*

**Table 5.** (*Continued*)

| | Austria | | Cyprus | | Finland | | Germany | | India | | North Macedonia | | Poland* | | USA | |
|---|---|---|---|---|---|---|---|---|---|---|---|---|---|---|---|---|
| | Est. (SE) | Std. Est. | Est. (SE) | Std. Est. | Est. (SE) | Std. Est. | Est. (SE) | Std. Est. | Est. (SE) | Std. Est. | Est. (SE) | Std. Est. | Est. (SE) | Std. Est. | Est. (SE) | Std. Est. |
| Procrastination | **0.08 (0.01)** | **0.06** | 0.02 (0.03) | 0.02 | 0.08 (0.06) | 0.07 | 0.05 (0.05) | 0.03 | 0.02 (0.02) | 0.02 | 0.02 (0.02) | 0.02 | **0.08 (0.04)** | **0.07** | -0.01 (0.01) | -0.01 |
| **Competence → Learning Motivation** | | | | | | | | | | | | | | | | |
| Engagement | **0.20 (0.01)** | **0.27** | 0.05 (0.06) | 0.06 | **0.18 (0.03)** | **0.30** | **0.23 (0.04)** | **0.32** | **0.11 (0.02)** | **0.18** | **0.13 (0.03)** | **0.17** | **0.12 (0.03)** | **0.18** | -0.03 (0.02) | -0.04 |
| Persistence | **0.14 (0.01)** | **0.18** | **0.12 (0.06)** | **0.15** | **0.12 (0.04)** | **0.16** | **0.12 (0.04)** | **0.15** | **0.10 (0.03)** | **0.13** | **0.18 (0.03)** | **0.21** | **0.09 (0.03)** | **0.13** | 0.00 (0.02) | 0.00 |
| Procrastination | **-0.13 (0.01)** | **-0.10** | 0.01 (0.06) | 0.01 | -0.10 (0.05) | -0.09 | **-0.16 (0.06)** | **-0.12** | **0.07 (0.03)** | **0.10** | -0.05 (0.05) | -0.04 | **-0.11 (0.04)** | **-0.09** | 0.01 (0.02) | 0.01 |
| **Autonomy → Positive Emotion** | | | | | | | | | | | | | | | | |
| Engagement | **-0.01 (0.00)** | **-0.01** | -0.01 (0.03) | -0.01 | -0.01 (0.01) | -0.02 | -0.02 (0.01) | -0.02 | 0.00 (0.00) | -0.01 | -0.01 (0.01) | -0.01 | 0.00 (0.01) | 0.00 | 0.00 (0.01) | 0.00 |
| Persistence | **-0.01 (0.00)** | **-0.02** | -0.03 (0.03) | -0.04 | -0.01 (0.01) | -0.02 | -0.02 (0.01) | -0.02 | -0.01 (0.01) | -0.01 | -0.02 (0.01) | -0.03 | 0.01 (0.01) | 0.01 | 0.01 (0.01) | 0.01 |
| Procrastination | **0.02 (0.00)** | **0.01** | 0.02 (0.04) | 0.02 | 0.02 (0.01) | 0.01 | 0.01 (0.01) | 0.01 | 0.01 (0.01) | 0.01 | 0.02 (0.02) | 0.02 | -0.01 (0.02) | -0.01 | 0.00 (0.01) | 0.00 |
| **Autonomy → Learning Motivation** | | | | | | | | | | | | | | | | |
| Engagement | **0.05 (0.00)** | **0.07** | 0.03 (0.04) | 0.03 | **0.05 (0.02)** | **0.08** | **0.04 (0.02)** | **0.05** | **0.02 (0.01)** | **0.03** | 0.02 (0.01) | 0.03 | **0.12 (0.04)** | **0.18** | 0.00 (0.01) | 0.00 |
| Persistence | **0.03 (0.00)** | **0.04** | 0.06 (0.05) | 0.07 | **0.04 (0.02)** | **0.04** | 0.02 (0.01) | 0.03 | 0.02 (0.01) | 0.02 | 0.03 (0.02) | 0.04 | **0.09 (0.03)** | **0.13** | 0.00 (0.00) | 0.00 |
| Procrastination | **-0.03 (0.00)** | **-0.02** | 0.00 (0.04) | 0.00 | -0.03 (0.02) | -0.02 | -0.03 (0.02) | -0.02 | 0.01 (0.01) | 0.02 | -0.01 (0.01) | -0.01 | **-0.12 (0.04)** | **-0.09** | 0.00 (0.01) | 0.00 |
| **Relatedness → Positive Emotion** | | | | | | | | | | | | | | | | |
| Engagement | **-0.02 (0.00)** | **-0.03** | -0.01 (0.02) | -0.01 | -0.02 (0.02) | -0.03 | **-0.03 (0.01)** | **-0.04** | -0.01 (0.01) | -0.01 | -0.01 (0.01) | -0.01 | -0.01 (0.01) | -0.03 | 0.01 (0.02) | 0.02 |
| Persistence | **-0.03 (0.00)** | **-0.03** | -0.03 (0.03) | -0.03 | -0.03 (0.02) | -0.03 | **-0.03 (0.01)** | **-0.03** | -0.01 (0.01) | -0.01 | -0.02 (0.01) | -0.02 | **-0.04 (0.02)** | **-0.06** | 0.04 (0.03) | 0.06 |
| Procrastination | **0.04 (0.00)** | **0.02** | 0.02 (0.03) | 0.01 | 0.04 (0.03) | 0.03 | 0.02 (0.02) | 0.01 | 0.01 (0.01) | 0.01 | 0.02 (0.01) | 0.01 | **0.05 (0.03)** | **0.04** | -0.02 (0.03) | -0.02 |
| **Relatedness → Learning Motivation** | | | | | | | | | | | | | | | | |
| Engagement | **-0.02 (0.00)** | **-0.02** | 0.01 (0.02) | 0.01 | -0.03 (0.02) | -0.03 | **-0.06 (0.02)** | **-0.07** | 0.00 (0.00) | 0.00 | -0.01 (0.01) | -0.02 | **0.05 (0.03)** | **0.08** | -0.01 (0.01) | -0.02 |
| Persistence | **-0.01 (0.00)** | **-0.01** | 0.02 (0.04) | 0.02 | -0.02 (0.01) | -0.02 | **-0.03 (0.01)** | **-0.03** | 0.00 (0.00) | 0.00 | -0.02 (0.01) | -0.02 | **0.04 (0.02)** | **0.06** | 0.00 (0.01) | 0.00 |
| Procrastination | **0.01 (0.00)** | **0.01** | 0.00 (0.02) | 0.00 | 0.01 (0.01) | 0.01 | **0.04 (0.02)** | **0.03** | 0.00 (0.00) | 0.00 | 0.01 (0.01) | 0.00 | **-0.05 (0.03)** | **-0.04** | 0.00 (0.01) | 0.00 |
| **Explained Variance (R²)** | | | | | | | | | | | | | | | | |
| Positive Emotion | .52 | | .21 | | .68 | | .43 | | .40 | | .36 | | .36 | | .34 | |
| Learning Motivation | .45 | | .35 | | .55 | | .47 | | .64 | | .51 | | .37 | | .28 | |
| Engagement | .36 | | .14 | | .36 | | .44 | | .37 | | .11 | | .25 | | .02 | |
| Persistence | .45 | | .66 | | .68 | | .50 | | .59 | | .48 | | .38 | | .04 | |

(*Continued*)

**Table 5.** (Continued)

| | Austria | | Cyprus | | Finland | | Germany | | India | | North Macedonia | | Poland* | | USA | |
|---|---|---|---|---|---|---|---|---|---|---|---|---|---|---|---|---|
| | Est. (SE) | Std. Est. | Est. (SE) | Std. Est. | Est. (SE) | Std. Est. | Est. (SE) | Std. Est. | Est. (SE) | Std. Est. | Est. (SE) | Std. Est. | Est. (SE) | Std. Est. | Est. (SE) | Std. Est. |
| Procrastination | .31 | | .25 | | .52 | | .35 | | .04 | | .17 | | .32 | | .01 | |

*Note*. Est. = Unstandardized parameter estimate; SE = Standard error; Std. Est. = Standardized estimate; Statistically significant results with $\alpha < .05$ are in boldface.

*As participants in Poland, due to technical issues, were only presented two items of the competence scale, the model for Poland was not analyzed within the multi-group model but separately.

persistence in Austria, Germany, North Macedonia, and Poland with small effect sizes, each ($\beta$ = -.11 to -.22). There were no statistically significant associations in other countries. A statistically significant effect of positive emotion on procrastination (H 2c) was detected in Austria and Poland. In contrast to the assumption, the effects were positive, but the effect sizes were small ($\beta$ = .10 and.15).

Consistent with Hypothesis 3a, the effect of intrinsic learning motivation on engagement was statistically significant in all countries except for Cyprus and the USA with medium to large effect sizes ($\beta$ = .25 to.60). Intrinsic learning motivation further predicted persistence (H 3b) in all countries except for the USA with small to medium effect sizes ($\beta$ = .17 to.41). The assumed negative effect of intrinsic learning motivation on procrastination (H 3c) was significant in Austria, Germany, and Poland with small to medium effect sizes ($\beta$ = -.16 to -.31). Another statistically significant, but positive association of intrinsic learning motivation and procrastination emerged in India. The effect size was, however, small ($\beta$ = .13).

**Indirect effects of the basic needs via positive emotion (H 4a-i).** The indirect effect of perceived competence on engagement via positive emotion (H 4a) was statistically significant in Austria and Germany. However, in contrast to the assumptions, the effects were both negative ($\beta$ = -.07 and -.11). The indirect effect of experienced autonomy on engagement (H 4b) was statistically significant in Austria. Contrary to the assumption, the effect was negative, but minor ($\beta$ = -.01). A similar pattern was identified for the indirect effect of perceived relatedness on engagement (H 4c). In contrast to the hypothesis, the identified effects in Austria ($\beta$ = -.03) and Germany ($\beta$ = -.04) were negative, but minor.

Indirect effects of perceived competence on persistence via positive emotion (H 4d) were identified in Austria, Germany, and Poland. In contrast to the assumptions, the effect sizes ($\beta$ = -.08 to -.11) were negative, but small. The indirect effect of experienced autonomy on persistence (H 4e) was statistically significant in Austria. Contrary to the assumption, the effect was negative, but minor ($\beta$ = -.02). A similar pattern arose for indirect effects of perceived relatedness on persistence (H 4f), which were only detected in Austria, Germany, and Poland. Contrary to the assumption, the effects were negative, but minor ($\beta$ = -.03 to -.06).

Indirect effects of perceived competence on procrastination via positive emotion (H 4g) were identified in Austria and Poland, only. Contrary to the assumption, the effects were positive ($\beta$ = .06 and.07), but small. Similar patterns arose for the indirect effects of perceived autonomy and relatedness on procrastination. Contrary to the assumption, positive, but minor effects were detected in Austria (H 4h, $\beta$ = .01), respectively in Austria and Poland (H 4i, $\beta$ = .02 and.04), only.

**Indirect effects of the basic needs via intrinsic learning motivation (H 5a-i).** The indirect effect of perceived competence on engagement via intrinsic learning motivation (H 5a) was statistically significant in all countries except for Cyprus and the USA. As hypothesized,

the identified effects were positive, with small to medium effect sizes ($\beta$ = .17 to.32). Positive indirect effects of perceived autonomy on engagement (H 5b) were identified in all countries but Cyprus, North Macedonia, and the USA. The effect sizes were, however, minor to small ($\beta$ = .03 to.18). An indirect positive, but small effect of experienced relatedness on engagement (H 5c) was identified in Poland only ($\beta$ = .08). Contradictory findings arose in Austria and Germany, where negative, but minor to small effects were detected ($\beta$ = -.02 and -.07).

Indirect effects of perceived competence on persistence via intrinsic learning motivation (H 5d) were identified in all countries except for the USA with small to medium effect sizes ($\beta$ = .13 to.21). Indirect effects of experienced autonomy on persistence (H 5e) were only statistically significant in Austria, Finland, and Poland, with minor to small effect sizes ($\beta$ = .04 to.13). An indirect positive, but small effect of perceived relatedness on persistence (H 5f) was identified in Poland only ($\beta$ = .06). Contradictory findings arose in Austria and Germany, where negative, but minor effects were detected ($\beta$ = -.01 and -.03).

Indirect, negative effects of experienced competence on procrastination via intrinsic learning motivation (H 5g) were detected in Austria, Germany, India, and Poland with small effect sizes ($\beta$ = -.09 to -.12). Indirect negative effects of perceived autonomy on procrastination (H 5h) were identified in Austria and Poland, only. However, the effect sizes were minor to small ($\beta$ = -.02 and -.09). As for the indirect effect of perceived relatedness on procrastination (H 5i), a result in line with our assumption was only identified in Poland, with a minor effect size, however ($\beta$ = -.04). Contrary to the assumption, positive, minor effect sizes were detected in Austria and Germany ($\beta$ = .01 and.03).

**Direct effects of the basic needs on learning behavior (H 6a-i).**    In line with Hypothesis 6a, perceived competence directly predicted engagement in Austria, Germany, and India, with medium effect sizes, each ($\beta$ = .24 to.38). Experienced autonomy was a direct predictor of engagement (H 6b) in India only, with a small effect size, however ($\beta$ = .12). Perceived relatedness had a direct effect on engagement (H 6c) in Austria, with a minor effect size ($\beta$ = .05).

In line with Hypothesis 6d, perceived competence directly predicted persistence in all countries, with medium to large effects ($\beta$ = .34 to.79), except for the USA who stood out of this pattern, as a small negative effect ($\beta$ = -.18) was detected. A positive direct effect of perceived autonomy on persistence (H 6e) was only detected in India with a small effect size ($\beta$ = .14). Direct positive effects of experienced relatedness on persistence (H 6f) were identified in Austria, North Macedonia, and the USA. The effect sizes were, however, small ($\beta$ = .07 to.17).

Assumed direct negative effects of perceived competence on procrastination (H 6g) emerged in all countries, except for India and the USA. The effects were medium to large ($\beta$ = -.36 and -.78). Significant direct effects of perceived autonomy on procrastination (H 6h) were identified in Austria, India, and North Macedonia. In contrast to the assumption, the effects were positive, but small ($\beta$ = .04 to.14). Finally, negative direct effects of experienced relatedness on procrastination (H 6i) were detected in Austria and North Macedonia with minor to small effect sizes ($\beta$ = -.04 and -.09).

**Comparison of patterns across countries (H 7).**    Overall, the study results show a number of comparable patterns across the countries of data collection. The most robust effect, which was identified in all countries, is the positive, direct effect of experienced competence on intrinsic learning motivation. Considering the confidence intervals of the effect sizes (see S3 Appendix), most of the intervals overlapped, indicating no statistically significant differences between the effects in Cyprus, Finland, Germany, India, and North Macedonia. The effect in Austria was higher than the effect in Finland, but did not differ statistically significantly from other countries, except for Poland and the USA. Poland and the USA, in turn, displayed statistically significantly lower effect sizes than all other countries, except for Cyprus.

Also, the direct effect of perceived competence on positive emotion proved rather robust and was identified in all countries, except for Cyprus. Statistically significant differences in the effect sizes emerged when comparing the USA with Austria, Finland, and Germany, who displayed higher effect sizes. Also, the effect in India was significantly lower than in the western European countries Austria, Germany, and Finland. Moreover, the effect in North Macedonia was significantly lower than in Austria and Germany.

Perceived competence further had relatively robust positive, direct effects on persistence, shown in all countries except for the USA, and negative direct effects on procrastination, shown in the six European countries. The confidence intervals for the effect sizes of perceived competence on persistence overlapped in all countries, except for the effect size in Finland, which was significantly higher than in Austria and Poland. A similar pattern arose for the effects of experienced competence on procrastination. All confidence intervals overlapped, except for the effect size in Finland, which was statistically significantly higher than in Austria, Cyprus, North Macedonia, and Poland. Regarding the effect of perceived competence on persistence, however, it should be noted, that one country stood out of the pattern: in the USA, experienced competence was negatively related to persistence.

Further robust direct effects of the basic psychological needs are effects of perceived autonomy on intrinsic learning motivation, shown in six countries, and experienced relatedness on positive emotion, shown in seven countries. Both effects did not differ statistically significantly between countries, except for the effect of perceived relatedness on positive emotion, which was significantly larger in the USA than in India.

Direct effects of the mediators, which have proven to be relatively robust, are effects of intrinsic learning motivation on engagement, shown in six countries, and persistence, shown in seven countries. In both cases, the hypothesized effect was not identified in the USA. The effect on engagement was further not detected in Cyprus. The confidence intervals for the effect sizes of intrinsic learning motivation on engagement mostly overlapped, except for the effect size in India which was statistically significantly smaller than in Austria, Finland, Germany, and Poland, as well as the effect size in North Macedonia, which was statistically significantly smaller than in Austria and Poland. For the effect of intrinsic learning motivation on persistence, all confidence intervals overlapped, indicating no statistically significant differences between effect sizes.

Indirect effects that have proven robust were the effects of perceived competence on engagement and persistence via intrinsic learning motivation. The indirect effect on engagement emerged in all countries except for Cyprus and the USA. Effect sizes did not differ statistically significantly, whereas the effect size in Austria was statistically significantly higher than in India. The indirect effect of perceived competence on persistence via intrinsic learning motivation did not differ statistically significantly between countries.

For further comparisons of effects that were identified in fewer than six of the eight countries of data collection, please refer to Table 5 and (S3 Appendix).

## Discussion

The present research aimed at identifying psychological characteristics that relate to adolescents' psychological well-being in terms of positive emotion and intrinsic learning motivation, and key characteristics of their learning behavior in a situation of unplanned distance education during the COVID-19 pandemic. Following SDT [11, 16], we assumed direct effects of basic psychological need satisfaction on learning behavior, as well as indirect effects, mediated via positive emotion and intrinsic learning motivation. Having collected data in altogether

eight countries from Europe, Asia, and North America, we expected to obtain comparable results across the countries of data collection.

Despite differences in single effects and effect sizes across countries, satisfaction of the basic need for competence was consistently found to relate to positive emotion and intrinsic learning motivation, and, in turn, active learning behavior in terms of engagement and persistence. The study results further highlight the role of perceived relatedness for positive emotion. The following section discusses the concrete findings along the hypotheses.

With regard to hypothesis set 1 (i.e., direct effects of basic need satisfaction on the mediators), the effects of experienced competence on positive emotion and intrinsic learning motivation proved empirically most robust across the countries of data collection. Statistically significant effects were identified in seven, respectively all eight countries. These effects were predominantly large. There were also small to medium effects of perceived autonomy on intrinsic learning motivation in six countries, and of experienced relatedness on positive emotion in seven countries. Positive, but small effects of perceived autonomy on positive emotion emerged in four countries. As for associations between perceived relatedness and intrinsic learning motivation, we detected results that partly contradicted our assumptions: While small, positive effects were identified in Poland and the USA, there were small, negative effects of experienced relatedness on intrinsic learning motivation in Austria, Germany and North Macedonia. The very small effect sizes, however, do not allow for concluding on the practical relevance of these effects. The findings with respect to hypothesis set 1 are mostly in line with SDT [11, 16], whereas more salient positive effects of perceived autonomy on the mediators were expected. A possible explanation for this could be the exceptional situation of the COVID-19 pandemic, where experienced autonomy might play a different role than under usual circumstances. Insofar, a high amount of perceived autonomy in an unstable situation might be experienced ambivalently, not necessarily leading to increased positive emotion and intrinsic learning motivation. So far, unfavorable effects of autonomy have primarily been investigated in the context of occupational psychology [82], but should also be considered in the context of (distance) education. The robust effect of perceived relatedness on positive emotion, however, is in line with previous studies pointing to the high relevance of social contacts for adolescent well-being [35, 36].

As for hypothesis sets 2 and 3 (i.e., direct effects of the mediators on learning behavior), positive effects of learning motivation on engagement and persistence proved relatively robust across countries, with predominantly medium to large effect sizes. Assumed negative effects of learning motivation on procrastination were only detected in three countries and assumed positive effects of positive emotion on engagement and persistence were not identified at all. In some countries, the results contradicted the hypotheses: A small, but positive association between intrinsic learning motivation and procrastination emerged in India, negative effects of positive emotion on engagement arose in Austria and Germany, and negative effects of positive emotion on persistence were shown in Austria, Germany, North Macedonia and Poland. This indicates that students in the respective countries, who displayed higher positive emotion were less active in learning, while students displaying lower positive emotion were more actively engaged and persistent when learning. A possible conclusion from this could be that active learning behavior might have served as a coping mechanism for students who displayed low positive emotion. This behavioral pattern has for example also been shown among young refugees [83]. On the other hand, low positive emotions could also be due to pressure of meeting assignment deadlines, attending online classes, or parental pressure to complete schoolwork, leading to more active learning behavior. Also in line with these possible explanations, there were a positive, small effects of positive emotion on procrastination in Austria and India.

Regarding hypothesis sets 4 and 5 (i.e., indirect effects of basic need satisfaction via positive emotion and intrinsic learning motivation), the indirect effect of perceived competence on engagement via intrinsic learning, shown in six countries, and persistence via intrinsic learning, shown in seven countries, proved most robust. Indirect effects of experienced autonomy on engagement via intrinsic learning motivation, shown in five countries, were also in line with our hypothesis. However, these relations could not be shown consistently and predominantly displayed minor effect sizes. Further indirect effects were only detected sporadically, with minor to small effect sizes, not allowing for conclusions to be drawn about their practical relevance.

As for hypothesis sets 6 (i.e., direct effects of basic need satisfaction on learning behavior), direct positive effects of experienced competence on persistence were shown in seven countries with medium to large effect sizes, whereas the USA stood out of this pattern, displaying a small negative effect. Assumed negative, medium to large effects of perceived competence on procrastination emerged in six countries. Other direct effects of the basic needs on learning behavior emerged only sporadically: Medium effect sizes emerged for the effect of experienced competence on engagement in three countries, while further direct effects of basic need satisfaction on learning behavior were only minor to small, not allowing for conclusions about their practical relevance. Given core postulates of SDT and previous empirical findings [49–51, 53], the minor effect of autonomy on learning behavior is a surprising finding.

Although the assumed hypotheses were not consistently confirmed, there is a high explanatory power for the dependent variables in the models. Accordingly, the models in the respective countries accounted for up to 68% of the explained variance for positive emotion, 64% for intrinsic learning motivation, 44% for engagement, 68% for persistence, and 52% for procrastination (see Table 5). However, it should be pointed out, that these values are much lower in some countries, and in some cases even close to zero. This means that the model set up was not able to explain the variance in the outcomes in all countries comparably (Hypothesis 7). The USA stands out in particular, with the lowest proportion of explained variance. In contrast, the highest proportions of explained variance emerged for Finland, followed by Germany, Austria, India, Poland, North Macedonia, and Cyprus. Whether these differences are due to country-specific characteristics, for instance with regard to the handling of the pandemic, our data does not allow for drawing final conclusions. This would require representative data to rule out the possibility of sample bias, and data on students' perception of the context and challenges bound to the pandemic to be better able to interpret the findings.

## Central conclusions for distance education in times of COVID-19

Despite the differences in findings, the role of experienced competence for positive emotion and intrinsic learning motivation, as well as active learning behavior in terms of engagement and persistence, is a central, consistent finding of the present multi-country study. This underlines the fundamental importance of students experiencing successes. Distance education in times of COVID-19 should therefore particularly focus on the provision of individualized learning opportunities that challenge students based on their strengths and weaknesses. In this respect, regular feedback, monitoring, and opportunities for improvement are considered essential to promote experienced competence among students (see also [38, 39]).

Whilst effects of perceived relatedness on variables related to intrinsic learning motivation and learning behavior were less salient, the results further highlight the role of experienced relatedness for positive emotion. Therefore, ways to promote feelings of relatedness even in the distance education context should increasingly be considered (see for example [84]). In this respect, we suggest using digital learning platforms to also encourage interactive group

work among students, to promote a feeling of learning together and to reflect on learning processes as a group in synchronous learning units (e.g., video conferences).

Contrary to our assumptions, effects of perceived autonomy on positive emotion and intrinsic learning motivation were less salient. This also applies to the assumed direct and indirect effects on variables associated with learning behavior. In some countries, experienced autonomy was even associated with higher levels of procrastination. This could be an indication, that experienced autonomy in times of learning under COVID-19 circumstances might have different characteristics than autonomy-supportive learning arrangements that take place in face-to-face classes. Accordingly, the great flexibility of distance education goes hand in hand with less structuring of daily routines, and fewer opportunities for teachers to intervene when learners experience difficulties in handling their autonomy. In line with previous studies pointing to the relevance of self-regulated learning for autonomous functioning [41], the role of learning organization and self-regulation in dealing with autonomy would certainly be a worthwhile approach for further research on learning in times of the COVID-19 pandemic.

## Study strengths, limitations, and future directions

The large sample size and the high explanatory power of our models can be considered as substantial strengths of the present research. Furthermore, we followed a multi-study approach and tested our assumptions in samples stemming from eight different countries. Several of the identified relations turned out to be relatively robust across countries. These results are further supported by the established measurement invariance.

Despite these noteworthy strengths, the present research is limited in some respects. First, this is a cross-sectional study, limiting the possibility for causal inferences. Second, data was collected online. This led to a self-selection of the samples. Also, the participating countries were self-selected and there were different prerequisites for recruiting the samples. While the Austrian sample comprises a very large sample due to a cooperation with the Federal Ministry of Education, Science and Research, the samples in the other countries were mostly recruited through individual contacts with educational stakeholders. As a consequence, the sample sizes between countries differed greatly, leading to inequivalent statistical power for detecting direct and indirect effects, as well as concluding on differences between effect sizes due to larger confidence intervals in countries with smaller samples. Finally, our study focused on the identification of consistent findings, rather than explaining different findings across the countries of data collection. This would require representative samples to rule out the possibility of sample bias, as well as data on the specific contexts and how they were perceived to be better able to interpret the findings.

Considering these limitations, we recommended future studies to incorporate further informants (e.g., teacher and parent ratings) to obtain a more comprehensive picture. It would also be profitable to explicitly consider the multilevel structure of the data (students in classes, classes in schools) to identify possible context effects. This could not be controlled for in this study, due to the anonymization of the data. More systematic approaches of sample recruitment should be considered to ensure a better representation of the population. Representative samples would further allow to take a closer look at the different effects identified across countries. Longitudinal studies could further substantiate the evidence for the large effects found. With regard to the implementation of distance education, it should be considered to evaluate various design options such as modality, pacing, instructional practices, role of assessments and feedback and to examine the extent to which they are suitable to enable basic psychological need satisfaction under COVID-19 circumstances. Finally, the role of learning organization

and self-regulation for dealing with autonomy could be a worthwhile approach for further research.

## Supporting information

**S1 Appendix. Bivariate latent correlations per country.** Each table section contains coefficients for two countries separated by the diagonal. Values in bold letters represent significant correlations at p < .05.
(PDF)

**S2 Appendix. Descriptive statistics for each country of data collection.**
(PDF)

**S3 Appendix. 95% Bias-corrected bootstrap confidence intervals for direct and indirect effects of the multi-group mediation model for each country of data collection.**
(PDF)

## Author Contributions

**Conceptualization:** Julia Holzer.

**Data curation:** Julia Holzer, Selma Korlat, Martin Mayerhofer.

**Formal analysis:** Julia Holzer.

**Funding acquisition:** Barbara Schober, Christiane Spiel, Marko Lüftenegger.

**Investigation:** Julia Holzer, Toumazis Toumazi, Katariina Salmela-Aro, Udo Käser, Anja Schultze-Krumbholz, Sebastian Wachs, Mukul Dabas, Suman Verma, Dean Iliev, Daniela Andonovska-Trajkovska, Piotr Plichta, Jacek Pyżalski, Natalia Walter, Justyna Michałek-Kwiecień, Aleksandra Lewandowska-Walter, Michelle F. Wright.

**Methodology:** Julia Holzer, Marko Lüftenegger.

**Project administration:** Julia Holzer, Selma Korlat, Christian Haider, Elisabeth Pelikan.

**Supervision:** Barbara Schober, Christiane Spiel, Marko Lüftenegger.

**Writing – original draft:** Julia Holzer.

**Writing – review & editing:** Selma Korlat, Christian Haider, Martin Mayerhofer, Elisabeth Pelikan, Barbara Schober, Christiane Spiel, Toumazis Toumazi, Katariina Salmela-Aro, Udo Käser, Anja Schultze-Krumbholz, Sebastian Wachs, Mukul Dabas, Suman Verma, Dean Iliev, Daniela Andonovska-Trajkovska, Piotr Plichta, Jacek Pyżalski, Natalia Walter, Justyna Michałek-Kwiecień, Aleksandra Lewandowska-Walter, Michelle F. Wright, Marko Lüftenegger.

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
