## [Decision Letter · Decision Letter 0]

23 Feb 2021

PONE-D-21-00976

Adolescent Well-being and Learning in Times of COVID-19 – A Multi-country Study of Basic Psychological Need Satisfaction, Learning Behavior, and the Mediating Roles of Positive Emotion and Intrinsic Motivation

PLOS ONE

Dear Dr. Holzer,

Thank you for submitting your manuscript to PLOS ONE. After careful consideration, we feel that it has merit but does not fully meet PLOS ONE’s publication criteria as it currently stands. Therefore, we invite you to submit a revised version of the manuscript that addresses the points raised during the review process.

We look forward to receiving your revised manuscript.

Kind regards,

Frantisek Sudzina

Academic Editor

PLOS ONE

Journal Requirements:

Reviewer's Responses to Questions

**Comments to the Author**

1. Is the manuscript technically sound, and do the data support the conclusions?

Reviewer #1: Yes

Reviewer #2: Yes

2. Has the statistical analysis been performed appropriately and rigorously? 

Reviewer #1: Yes

Reviewer #2: Yes

3. Have the authors made all data underlying the findings in their manuscript fully available?

Reviewer #1: Yes

Reviewer #2: Yes

4. Is the manuscript presented in an intelligible fashion and written in standard English?

Reviewer #1: Yes

Reviewer #2: Yes

5. Review Comments to the Author

Reviewer #1: I have read the paper on adolescent well-being and learning in times of COVID-19 with great interest. I found the manuscript to be very well developed and to address a highly important and relevant topic. Furthermore, I also found the manuscript to be very well written, clearly argued for, and well discussed. Notable strengths of the manuscript entail its large data-set, the rigorous testing of central assumptions (such as measurement invariance), and the detailed description of the findings. I believe that the manuscript will be very well received by readers of PLOS ONE and congratulate the authors to an important contribution to the literature. I only have a very few points that the authors may want to consider when revising their manuscript.

1. Page 5: Advantages of autonomy-supportive learning environments: These also require the students to have the necessary SRL skills (which, they frequently, do not have). Therefore, learning environments should critically consider such aspects, e.g. by prompting students to employ adequate learning strategies. The authors may want to consider this point by relativizing their argument.

2. Page 6: “To the best of our knowledge …” + next sentence: These two sentences seem to contradict each other. Please rephrase to clarify this.

3. Generally, I recommend to write about “competence satisfaction” instead of solely “competence” throughout the manuscript (also regarding autonomy and relatedness of course) in order to be clear in writing and avoid misunderstandings.

4. I was wondering how the students were recruited, particularly with regard to them possibly being from different classrooms. If they are from different classrooms, it might be helpful to provide information on the ICCs of the assessed constructs to make sure that there are no substantial amounts of shared variance that could cause problems with model estimation.

5. The CFI and RMSEA cut-offs by Chen seem very adequate, especially as Chi²-differnece tests would be biased by the large sample sizes. The cut-offs could be used in an even more detailed manner. Chen recommends partly different cut-offs for the different steps of measurement invariance testing.

6. Note of Table 4: Were the items or the answers from Poland not included? Please clarify what “they” refers to accordingly.

7. Differences between different countries in model path estimates: As some of the countries have much smaller sample size than others, it is not surprising that some parameter estimates are not significant, despite being descriptively in the same order of magnitude as parameter estimates from larger countries (e.g. second parameter estimate for Austria and Cyprus). Power analyses could consequently be included to make a stronger points that this does not imply that there are meaningful differences between the different countries.

8. I was surprised that the differences in parameter estimates were based on the confidence intervals. The most straightforward approach to me would have been to extend the logic of measurement invariance testing and restrict the parameters to be equal across the different groups. If full invariance is not attested, restrictions can be loosened until partial invariance is reached. The authors may wish to consider this or to provide a stronger rationale for the approach that they used (possible, due to the large amount of countries, the first approach that I suggested by not be so sensible after all).

Reviewer #2: This is an interesting and timely study which examines adolescents’ well-being during Covid-19 pandemic across eight countries using a large sample size which further strengthens findings and generalizability of the findings. Here are my comments and suggestions for the authors that I hope they find helpful:

1- Except for Cyprus which has an age range of 12-17, other countries have the maximum age of for example, 20 or 21. I’m not sure whether a person whose age is 21 is attending a secondary school. This might be true, but at least some clarification might be necessary here to clarify the different contexts.

2- Did authors control for the age? As it can be seen from Table 1, the age range for some countries is between 10-19 or 10-21. It seems this age range includes more than adolescents and it can affect the analyses.

3- Authors stated that they used bootstrapping for testing significance. As far as I now, MLR cannot produce bootstrapping confidence intervals. Can authors explain how they obtained bootstrapped confidence intervals with MLR? Did they run the models again?

4- It might be helpful to include ∆CFI and ∆RMSEA in Table 4.

5- Concerning Table5, it seems authors have made significant relations bold, but this has not been mentioned in the note of Table 5.

6- Regarding H7, authors state that overlapping confidence intervals did not differ from one another. Can authors clarify what does exactly “overlapping confidence intervals” mean?

7- While the text is generally well-written, there are some linguistic issues with regard to the text which need to be revised. Here are some examples:

8- “data were collected in altogether eight countries in Europe, Asia and North America” -> I think this sentence needs to be revised. My suggestion: “data were collected in altogether FROM eight countries in Europe, Asia and North America”.

9- P.10: “Before being forwarded to the items, …” -> I can’t understand this phrase. Do authors mean “Before answering the items,…”?

10- “The statistically significant effects that were identified in seven, respectively all eight countries, were predominantly large.” -> this sentence needs to be revised.

11- “Unfavorable effects of autonomy have so far primarily been drawn attention to in the context of occupational psychology” -> this sentence needs to be revised.

6. PLOS authors have the option to publish the peer review history of their article (what does this mean?). If published, this will include your full peer review and any attached files.

Reviewer #1: No

Reviewer #2: **Yes: **Gholam Hassan Khajavy

---

## [Author Response · Author response to Decision Letter 0]

21 Apr 2021

We wish to thank the editor and the reviewers for their positive assessment of our manuscript and for the feedback, which helped us to further improve our work. This is much appreciated. We address the reviewers’ comments in a seperate file named "Response to Reviewers". For each of the comments, we present our answers in detail and point out the corresponding changes in the revised manuscript. From our perspective, the quality of the paper has been significantly improved. We thank the editor and the reviewers for their time and attention devoted to this manuscript and hope to obtain a favorable review.

---

## [Editor Report · Decision Letter 1]

26 Apr 2021

Adolescent Well-being and Learning in Times of COVID-19 – A Multi-country Study of Basic Psychological Need Satisfaction, Learning Behavior, and the Mediating Roles of Positive Emotion and Intrinsic Motivation

PONE-D-21-00976R1

Dear Dr. Holzer,

We’re pleased to inform you that your manuscript has been judged scientifically suitable for publication and will be formally accepted for publication once it meets all outstanding technical requirements.

Kind regards,

Frantisek Sudzina

Academic Editor

PLOS ONE
---

## [Editor Report · Acceptance letter]

4 May 2021

PONE-D-21-00976R1 

Adolescent Well-being and Learning in Times of COVID-19 – A Multi-country Study of Basic Psychological Need Satisfaction, Learning Behavior, and the Mediating Roles of Positive Emotion and Intrinsic Motivation 

Dear Dr. Holzer:

I'm pleased to inform you that your manuscript has been deemed suitable for publication in PLOS ONE. Congratulations! Your manuscript is now with our production department. 

Kind regards, 

on behalf of

Dr. Frantisek Sudzina 

Academic Editor

PLOS ONE